# Norovirus Intestinal Infection and Lewy Body Disease in an Older Patient with Acute Cognitive Impairment

**DOI:** 10.3390/ijms23158376

**Published:** 2022-07-29

**Authors:** Manuel Moreno-Valladares, Veronica Moncho-Amor, Iraide Bernal-Simon, Eñaut Agirre-Iturrioz, María Álvarez-Satta, Ander Matheu

**Affiliations:** 1Pathology Department, Donostia University Hospital, Osakidetza Basque Health Service, 20014 San Sebastian, Spain; iraide.bernalsimon@osakidetza.eus (I.B.-S.); enaut.agirreiturrioz@osakidetza.eus (E.A.-I.); 2Group of Cellular Oncology, Biodonostia Health Research Institute, 20014 San Sebastian, Spain; veronica.moncho@biodonostia.org (V.M.-A.); maria.alvarez@biodonostia.org (M.Á.-S.); ander.matheu@biodonostia.org (A.M.); 3CIBER on Frailty and Healthy Aging (CIBERfes), Institute of Health Carlos III, 28029 Madrid, Spain; 4IKERBASQUE, Basque Foundation for Science, 48011 Bilbao, Spain

**Keywords:** alpha-synuclein, Lewy bodies, norovirus, dementia with Lewy bodies

## Abstract

We present a case report on an older woman with unspecific symptoms and predominant long-term gastrointestinal disturbances, acute overall health deterioration with loss of autonomy for daily activities, and cognitive impairment. Autopsy revealed the presence of alpha-synuclein deposits spread into intestinal mucosa lesions, enteric plexuses, pelvic and retroperitoneal nerves and ganglia, and other organs as well as Lewy pathology in the central nervous system (CNS). Moreover, we isolated norovirus from the patient, indicating active infection in the colon and detected colocalization of norovirus and alpha-synuclein in different regions of the patient’s brain. In view of this, we report a concomitant norovirus infection with synthesis of alpha-synuclein in the gastrointestinal mucosa and Lewy pathology in the CNS, which might support Braak’s hypothesis about the pathogenic mechanisms underlying synucleinopathies.

## 1. Introduction

Lewy bodies (LBs) are intracellular accumulations of alpha-synuclein protein that have been observed in several neurodegenerative disorders for a long time, including Parkinson´s disease (PD). Nonetheless, it was not until the 1990s that alpha-synuclein protein was identified as the main component of these aggregates [1,2]. LBs are divided into classical and cortical types. Classical LBs are spherical (8–30 µM), intracytoplasmic eosinophilic inclusions with narrow pale halos that can be observed in neuron cytoplasm and neuronal prolongations (intraneuritic LBs). On the other hand, cortical LBs are eosinophilic rounded, reniform or angular cytoplasmic inclusions, without halos [3]. Dystrophic Lewy neurites (diffuse deposits of alpha-synuclein in neuronal prolongations) and pale bodies (LB precursors observed as an accumulation of pale eosinophilic granule material) have been also described.

Alpha-synuclein is a 140-amino-acid protein with three different isoforms encoded by genes localized in chromosomes 4 (α), 5 (β), and 10 (γ). This protein harbors the KTKEGV consensus motif to bind and interact with lipid membranes. Alpha-synuclein is located at presynaptic terminals [4] where it is mainly involved in synapsis homeostasis [5]. Alpha-synuclein related toxicity is not well understood, but some studies point out that it interacts with cytochrome c oxidase, leading to neuronal death due to energetic deficit and oxidative stress [3,6]. Furthermore, LBs could represent residual accumulations of pathological mechanisms that are poorly characterized, such as other neurodegeneration-related inclusions [7,8]. In this case, dysfunction of the ubiquitin/proteasome system and autophagic-lysosomal pathway could lead to toxic protein aggregates´ formation [9].

The initial steps of LB formation consist of the emergence of alpha-synuclein oligomers that progressively aggregate in larger forms that are deposited as dystrophic neurites and LBs [10]. The alpha-synuclein accumulation may start in terminal axons leading to synapse disruption and neuronal death [11]. Axonal transport disruption can also be triggered by alpha-synuclein aggregates [12]. Alpha-synuclein dissemination within nerve fibers can be anterograde or retrograde [13], but some experiments indicate its deposit occurs first in axons and then in neuron bodies and dendrites [14]. This suggests a predominant retrograde transport at initial steps mainly due to alpha-synuclein accumulation in presynaptic terminals.

In recent decades, the formation of alpha-synuclein aggregates has been proposed to start in the gastrointestinal tract [15,16] from where they would be disseminated to the central nervous system (CNS) in a prion-like way [17,18]. Alpha-synuclein seems to play a protective role in the gastrointestinal tract for pathogen-carrier endocytic vesicles, such as viruses, in order to block infection propagation [19,20]. This evidence is closely linked to Braak´s hypothesis, which postulates that PD, the most common form of synucleinopathy, is originated from gastrointestinal damage [21]. In this sense, a recent report in pediatric patients, including intestinal allograft recipients, with gastrointestinal inflammation due to norovirus has shown the presence of alpha-synuclein secretion in the gut mucosa in response to norovirus infection [22]. Remarkably, this secretion positively correlates with the inflammation degree and persists during active viral replication. Norovirus is an RNA virus from the Caliciviridae family, which can cause mild to severe self-limited acute digestive distress as well as chronic infections in immunosuppressed patients [23]. This virus can infect neuroendocrine cells of gastrointestinal mucosa innervated by the sympathetic nervous system, leading to neurotransmitter and vasoactive substances release, which is responsible for the patient´s symptoms [24]. These infected neuroendocrine cells may represent the entry door to the CNS for norovirus and secreted alpha-synuclein [25,26]. 

Dissemination of alpha-synuclein aggregates through the CNS was described in sporadic PD by Braak and colleagues, who established six consecutive and accumulative stages [27]. In each stage, the presence of alpha-synuclein aggregates is assessed by the number of inclusions per 200× magnification field of view. Later, alternative methodologies by McKeith, Leverenz, and Beach were developed, all of them based on semi-quantitative scoring of protein aggregation and their anatomical distribution. Consensus criteria have recently been published, trying to overcome the inter-rater variability observed when using the different semi-quantitative staging systems mentioned above [28]. The LP consensus criteria (LPC) are based on the McKeith system, although they introduce a dichotomous variable (presence or absence of Lewy neurites or LBs) and include two new stages (amygdala-predominant and olfactory-only stages).

## 2. Case Presentation

We present the case of a 76-year-old woman with a history of high blood pressure, dyslipidemia, L4-L5 herniated disk surgery with ascendant fragment, and depression. Her mother was previously diagnosed with sporadic PD, but genetic analysis was not performed.

Between 2016 and 2018, the patient reported intestinal motility disorder episodes, with a 10 kg weight loss. Computerized tomography was normal, and colonoscopy found a tubular adenoma in the sigmoid colon without evidence of norovirus. In 2020, she returned to medical consultation with the same clinical picture. In this case, norovirus was isolated after stool culture, and the patient started a treatment with plantain. Three months later, she was admitted to the hospital due to overall health deterioration and disorientation, but she did not present focal neurologic deficits, resting or postural tremor, or changes in facial expression or visual fixation. PCR for coronavirus was negative. Her relatives reported progressive cognitive impairment with alteration of executive (little initiative, no capacity to develop activities without supervision or stimulus, sedentary behavior, gait deterioration) and memory functions (short-term memory loss and a tendency to remember past events). These cognitive alterations became worse during Spanish COVID-19 confinement (March–June 2020), limiting her ability to carry out daily activities by herself.

Just 10 days before admission, a single history of fall with no apparent head trauma was reported. A few hours after admission, she developed low-grade fever, probably from respiratory origin, and was treated with amoxicillin/clavulanic acid. The next morning, the patient was found without spontaneous breathing, with perioral bilious content and acute diarrhea, and finally died. Blood cultures were negative. 

## 3. Thoracic and Abdominal Organ Findings

An irregular ulcer of 5 cm was observed in the transition from the sigmoid colon to the rectum (Figure 1A). At microscopic level, mucosa and submucosa layers were affected, displaying an abundant, chronic inflammatory exudate as well as fibrosis signs (Figure 1B,C). Neurons from the ganglia of Meissner´s plexus (Figure 1D) and Auerbach´s plexus (Figure 1E) in the enteric nervous system (ENS) were strongly stained for alpha-synuclein. In line with previous studies [29], positive cells were also found in smooth muscle fibers. In addition, a pattern of alpha-synuclein diffuse cytoplasmic staining was observed in peripheral nerves that innervate the mesocolon (Figure 1F) and other pelvic regions, as well as in cardiac nerves (Figure 1G) and nerves that supply adrenal glands (Figure 1H,I), where LBs were found in adrenal medulla cells.

## 4. Central Nervous System Findings

The entire brain weighed 1.096 kg and presented an atrophic aspect. This atrophy was particularly striking at the dorsal region of the bilateral primary somatosensory cortex extending through the association cortex (Figure 2A,B). At the microscopic level, the atrophic somatosensory cortex displayed a six-layer structure of normal aspect without apparent neuronal loss. The white matter was clearly atrophied but no necrotic, inflammatory, or hemorrhagic lesions were observed (Figure 2C,D). However, there was some degree of gliosis at the surface level (cortical layers I and II) and deep level (cortical-subcortical transition area) (Figure 2E). We observed LBs in this cortical region, more frequently in layer IV (Figure 2F). The midbrain exhibited pallor of the substantia nigra, especially in the right side (Figure 2G). Classical LBs in pars compacta neurons (Figure 2H) were identified by hematoxylin-eosin staining and alpha-synuclein immunohistochemistry (Figure 2I) as well as dystrophic neurites (Figure 2J). 

Regarding brain regions without macroscopic alterations, we found Lewy neurites and classical LBs on both the dorsal vagal nucleus and the solitary nucleus (Figure 3A–C,G,H). Lewy pathology was accompanied by abnormal neurofilament deposition (Figure 3D,I), little Tau protein aggregates (Figure 3F), and scarce microglia proliferation in the affected neurons (Figure 3E,J). LBs were also observed in the reticular formation of the medulla oblongata but not in the olivary nucleus. This contrasts with the cases presenting multiple system atrophy [30]. In the locus coeruleus, we identified multiple classical LBs as we described in the substantia nigra (Figure 4A). LBs were especially abundant in the nucleus basalis, together with Lewy neurites (Figure 4B,C), but they were also present in the hypothalamus and more scarcely in the thalamus (Figure 4D–F). Remarkably, we observed distortion of gracile fasciculus and posterior columns of the spinal cord, possibly due to the presence of dystrophic fibers with abnormal neurofilament or alpha-synuclein deposits. In addition, isolated classical LBs in the spinal trigeminal nucleus were identified (Figure 4G,H). Classical and cortical LBs appeared in the neuropil of all brain cortex areas (Figure 4I), predominantly in deep layers (IV–V). We also observed LBs in deep brain structures, such as the striatum (Figure 4J,K), amygdala (Figure 4L), and hippocampus. Focusing on the hippocampal region, sporadic LBs in the deep entorhinal region extending to the perirhinal cortex were described. Moreover, the highest density of abnormal deposits was found in the subiculum (Figure 4M), the CA1 and CA2 subfields. Both LBs and dystrophic neurites were described in the CA1-3 subfields, mainly in the stratum radiatum (CA1) and sporadically in the stratum oriens. Regarding the CA3 region, a diffuse granular deposit that covered the soma and primary dendritic plasma membrane was noted but not LBs or intraneuritic deposits (Figure 4N,O). We identified Tau protein aggregates as neurofibrillary tangles together with dystrophic neurites in the entorhinal cortex, subiculum, and the CA1 subfield (Figure 4P). No beta-amyloid deposits were observed in this brain region (Figure 4Q). Finally, Lewy pathology signs (LBs and dystrophic neurites) were also described in the olfactory bulb (Figure 4R). Interestingly, the presence of norovirus was detected in the myenteric plexus (Figure 5A) and intestine (Figure 5B). Moreover, norovirus capsid protein VP1 was detected in different regions of the brain, including the dorsal vagus nucleus and the hippocampus (Figure 5C,E), where it colocalized with alpha-synuclein (Figure 5D,F).

## 5. Discussion and Conclusions

We reported a case of a 76-year-old woman with an atrophic brain showing extensive signs of synucleinopathy, i.e., classical, cortical, and intraneuritic LBs, as well as dystrophic neurites throughout her brain. This distribution pattern corresponds to the “Neocortical Lewy pathology” category according to the LPC system [31] (Table 1). These criteria were designed to reduce the interobserver variability in the semi-quantitative analysis of Lewy pathology concerning extension and number of lesions. In this sense, synucleinopathies, as a clinical entity, are currently being revised, as protein aggregates other than alpha-synuclein can be found in this group of pathologies. In turn, other neurodegenerative diseases can exhibit alpha-synuclein deposits, e.g., Alzheimer´s disease with LBs. The causal link between protein aggregates and neurodegenerative processes is still not clear [32]; indeed, age-associated neurodegenerative diseases are not mutually exclusive [28]. In our case, Lewy disease was accompanied by tauopathy, but no beta-amyloid deposits were observed. This led us to exclude a diagnosis of Alzheimer´s disease with LBs. 

The great affection of the hippocampal region could explain the patient´s memory problems. Moreover, the presence of dystrophic neurites in peripheral fibers that innervate the heart and adrenal glands, among others, could indicate that vegetative alterations are behind the fall recorded in her medical history. This kind of affection is often observed in both pure vegetative forms of Lewy disease [33] and multiple system atrophy [30], but we did not find alterations in the olivary nucleus nor alpha-synuclein deposits in oligodendrocytes, which constitute hallmarks of this last disease [34]. We also described reduced pigmentation, presence of neurons with LBs, and individual melanin granules in the substantia nigra that suggest neuronal loss. However, our patient did not refer symptoms consistent with PD, such as resting tremor and cogwheel rigidity. Conversely, bradykinesia, gastrointestinal motility disturbances, and falls possibly due to orthostatic hypotension were reported. Concerning the diagnosis of dementia with LBs (DLB), the former symptoms together with cognitive impairment were present in our patient but not well-formed visual hallucinations or Parkinsonism. DLB can be categorized into two groups: possible and probable, taking into account the presence of main and secondary criteria as well as biomarker determination [35], which was not performed in our case. Despite this fact, the patient´s autopsy led us to establish a differential diagnosis of DLB as the main underlying pathology. 

The widespread distribution and abundance of LBs and Lewy neurites in the brain, ENS, intestinal mucosa, peripheral nerves, and other organs (e.g., adrenal glands) of our patient is mostly coincident with that reported by other works in patients differentially diagnosed with DLB [36,37,38]. Although some of them specifically evaluated the phosphorylated form of alpha-synuclein (present only in pathological deposits), our findings corroborate the distribution previously described for this group of patients.

The co-occurrence of norovirus intestinal infection and DLB raises the possibility that viral infection might have contributed to the patient´s neurodegeneration since norovirus infection has been linked to intestinal alpha-synuclein secretion [22]. Our finding of alpha-synuclein aggregates in the patient´s intestinal mucosa and ENS could support Braak´s hypothesis of the gastrointestinal origin of alpha-synuclein deposits in the brain [21,25]. However, it cannot be excluded that there might be alternative ways via which the disease could develop or via which infection could impact the CNS. Additional studies have detected enteric alpha-synuclein in healthy individuals, raising concerns regarding the specificity of Braak´s hypothesis [39,40]. In our study, we found an ulcerated lesion, with abundant chronic inflammation and fibrosis, which indicates a chronic evolution, and detected norovirus particles associated with alpha-synuclein in the LBs by immunological techniques. The virus was detected in the feces at least 3 months before admission to the hospital, so it persisted in the area of the lesion for at least this period. Since norovirus intestinal infection is associated with presence of alpha-synuclein [22] and it positively correlates with the persistence of the virus, the detected alpha-synuclein in response to norovirus infection could migrate through the autonomic nervous system from the intestinal mucosa to the CNS and contribute to the development of the disease in susceptible patients. It is not well understood if alpha-synuclein aggregates trigger the neurodegenerative process or if they are a collateral effect or even a protective mechanism against protein homeostasis loss due to a protein degradation system dysfunction. In this respect, we observed positive staining for ubiquitin and other proteins involved in cellular stress response, such as p38 mitogen-activated protein kinase (p38MAPK) in LBs. Furthermore, p38 MAPK was also found in cells without alpha-synuclein aggregates. 

In conclusion, we reported a case of a 76-year-old woman suffering from gastrointestinal motility disturbances for years, who developed subacute symptoms of health deterioration in the general condition and progressive cognitive and functional decline. Fecal cultures for norovirus were positive, and there was colocalization of norovirus VP1 protein and alpha-synuclein deposits in the ENS and CNS. Although it could be plausible that the LB pathology was primarily of the CNS and was accelerated by the intercurrence of viral infection by norovirus, these results support Braak’s hypothesis concerning the pathogenic mechanisms underlying synucleinopathies.

## 6. Material and Methods

### 6.1. Samples and Clinical Data

Human tissue samples were collected from an autopsy conducted at Donostia University Hospital (Spain). Detailed clinical information of the patient was recorded at the time of admission through a clinical interview, including clinical exploration, anamnesis, and follow-up. Inquiries about the patient´s health condition were made to immediate relatives, both before and after the patient´s death.

The patient’s caregivers read and signed an informed consent after obtaining detailed information about the research. This study adhered to the tenets of the Declaration of Helsinki by the World Medical Association regarding human experimentation.

### 6.2. Tissue Processing, Immunohistochemistry, and Immunofluorescence

Briefly, three body cavities were opened and the organ block was eviscerated. After that, anatomical dissection was performed and organs were kept in a fixative solution (4% paraformaldehyde) for a period of 24–48 h. Fixed organs were examined at macroscopic level, then paraffin-embedded (LOGOS Microwave Hybrid Tissue Processor, Milestone) and sectioned in 4 µM (thoracic and abdominal organs) or 6 µM (brain) sections with a Leica RM 2155 rotary microtome. Brain samples were dried for 24 h prior to staining.

Immunohistochemistry was performed following standard procedures and manufacturer´s instructions on the Roche Ventana Benchmark ULTRA System. The following primary antibodies were used: alpha-synuclein (BioLegend, Ref.: 807701), anti-β-Amyloid, 1-42 antibody (BioLegend, Ref.: 805501, Clone: 12F4), phospho-tau (ThermoFisher, Ref.: MN1020, Clone: AT8), NeuN (Millipore, Ref.: MAB377, Clone: A60), neurofilament (Roche, Ref.: 760-2661), ubiquitin (BioLegend, Ref.: 646301), Iba1 (Santa Cruz Biotechnology, Ref.: sc-32725), GFAP (Cell Marque, Ref.: 760-4345), and p38 MAPK (Cell Signaling Ref.: 46315). Hematoxylin-eosin staining and Masson’s trichrome staining were carried out under standard procedures on the Roche Ventana HE600 system. All stained sections were visualized with a Leica DM2500 LED light microscope. 

Immunofluorescence was performed in formalin-fixed tissue samples. Paraffin-embedded tissue sections were deparaffinized in xylene, rehydrated in a series of graded alcohols, and then heated in citrate buffer during 1 h for antigen retrieval. Tissues were permeabilized with 0.5% Triton X-100 (PBS-T; Sigma-Aldrich, St. Louis, MO, USA) and blocked for 1 h with 1% bovine serum albumin and 5% goat-serum (Sigma-Aldrich) in PBS-T. Sections were incubated at 4 °C overnight with the following primary antibodies: Anti-Norovirus Capsid protein VP1 (Abcam, Ref.: ab92976) and alpha-synuclein (BioLegend, Ref.: 807701). Sections were washed in PBS-T and then incubated for 1 h at room temperature with the corresponding anti-mouse or anti-rabbit secondary antibody conjugated to Alexa Fluor 555 or 647 in 10% blocking solution with 1 μM 4′,6-diamino-2-phenylindole (DAPI). Finally, sections were washed with PBS-T and mounted using ProLong™ Gold Antifade Mountant (Invitrogen™, Waltham, MA, USA). Images were acquired using a Zeiss LSM 900 confocal microscope. Settings were established during the initial acquisition. 

### 6.3. Data Analysis

We performed an exhaustive analysis of all microscopic lesions, which were scanned with Virtuoso v.5.6 software (Ventana Medical Systems, Roche, Basel, Switzerland). Alpha-synuclein deposition analysis was done using the new LPC consensus criteria mentioned above in order to assess Lewy pathology signs in the different anatomic regions.

## Figures and Tables

**Figure 1 ijms-23-08376-f001:**
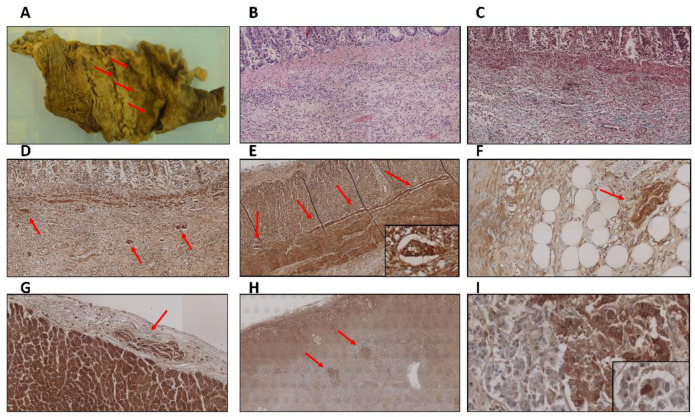
(**A**) Superficial ulcerated lesion with irregular borders in the colon mucosa. (**B**) Hematoxylin-eosin staining showing the colon mucosa with presence of abundant chronic inflammation in the submucosa layer (4×). (**C**) Masson´s trichrome staining revealed fibrosis signs in the same region showed in (**B**) (4×). (**D**–**F**) Diffuse cytoplasmic deposits of alpha-synuclein in neurons from the ganglia of Meissner´s plexus (**D**) (4×), Auerbach´s plexus (**E**) (4× and zoomed area at 40×), and peripheral nerves that supply the mesocolon (**F**) (4×). (**G**–**I**) Alpha-synuclein aggregates in peripheral nerves innervating the heart (**G**) (4×) and adrenal glands (**H**) (4×), where Lewy bodies were also observed (**I**) (4× and zoomed area at 40×).

**Figure 2 ijms-23-08376-f002:**
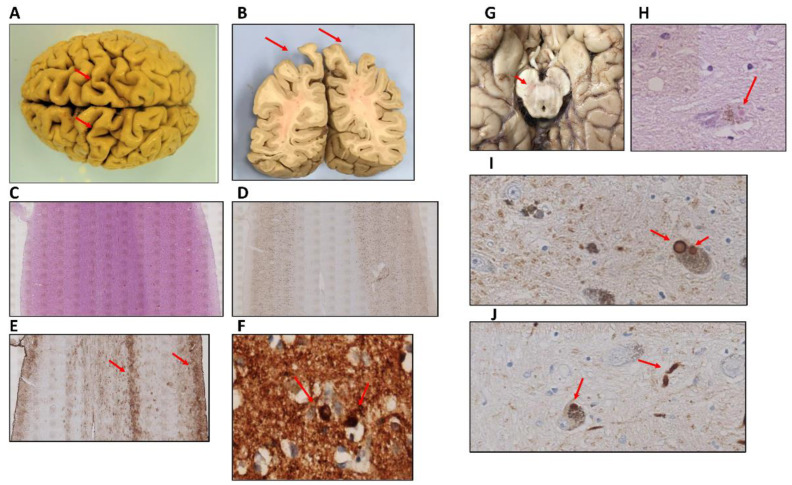
(**A**,**B**) Sagittal-oriented brain with atrophy at both the dorsal region of the bilateral primary somatosensory cortex (**A**) and the association cortex in the superior parietal lobule (**B**). (**C**) Hematoxylin-eosin staining of primary somatosensory cortex showing a marked white matter reduction but not evidence of necrotic or inflammatory areas (2×). (**D**,**E**) Immunohistochemistry for NeuN (**D**) (2×) and GFAP (**E**) (2×) markers revealed no significant neuronal loss but some degree of gliosis in the primary somatosensory cortex. (**F**) Alpha-synuclein aggregates in neurons from IV layer of somatosensory cortex (40×). (**G**–**J**) Substantia nigra was less pigmented (**G**) and presented neurons with classical Lewy bodies (**H**) (40×), multiple Lewy bodies (**I**) (40×), and dystrophic neurites (**J**), (40×).

**Figure 3 ijms-23-08376-f003:**
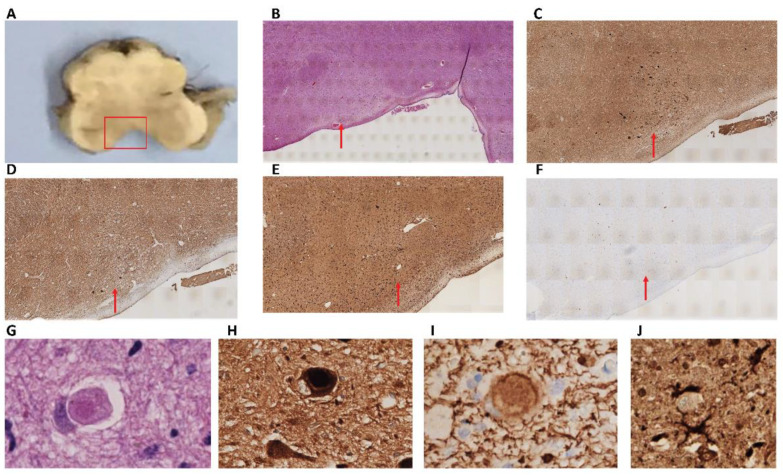
(**A**) Anatomical location of the dorsal vagal nucleus and the solitary nucleus in the medulla oblongata. (**B**) Hematoxylin-eosin staining did not reveal inflammatory, necrotic, or hemorrhagic lesions (2×). (**C**) Abundant alpha-synuclein deposits as Lewy bodies and dystrophic neurites (2×). (**D**) Neurofilament deposition was also observed in this anatomic region (2×). (**E**) Scarce microglia proliferation using Iba1 as specific marker (2×). (**F**) Tau-positive dystrophic neurites were also identified in medulla oblongata (2×). (**G**) Intracytoplasmic Lewy body stained by hematoxylin-eosin (80×). (**H**) Intracytoplasmic Lewy body marked for alpha-synuclein (80×). (**I**) Cytoplasmic deposits of neurofilament (80×). (**J**) Microglia (marked with Iba1) contacting a neuron (40×).

**Figure 4 ijms-23-08376-f004:**
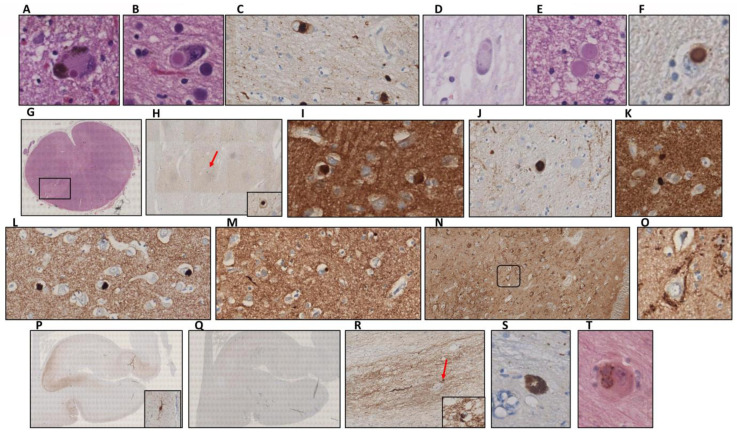
(**A,B**) Hematoxylin-eosin staining showing Lewy bodies in the locus coeruleus (**A**) and the nucleus basalis (**B**) at 40×. (**C**) Lewy bodies in the nucleus basalis marked for alpha-synuclein (40×). (**D**,**E**) Classical (**D**) and intraneuritic (**E**) Lewy bodies in the hypothalamus after hematoxylin-eosin staining (40×). (**F**) Lewy body in the hypothalamus marked for alpha-synuclein (40×). (**G**) Spinal trigeminal nucleus with the selected region for further study in a square. (**H**) Lewy body marked with alpha-synuclein from the region described in (**G**) (4× and zoomed area at 40×). (**I**) Lewy bodies in the cingulate cortex (40×). (**J**,**K**) Lewy bodies located in the striatum: putamen (**J**) and caudate (**K**) nuclei at 40×. (**L**) Lewy bodies in the amygdala (40×). (**M**) Lewy bodies in the subiculum (40×). (**N**) Alpha-synuclein aggregates on the neuronal soma membranes in the hippocampal CA3 subfield (10×). (**O**) Zoomed area of the region highlighted in (**N**) (40×). (**P**) Hippocampal tau deposits (2×) with a neurofibrillary tangle in a zoomed square. (**Q**) Absence of beta-amyloid deposition in the hippocampus (2×). (**R**) Dystrophic neurites in the olfactory bulb (10×) (Lewy body in a square). (**S**) Ubiquitin deposit in a Lewy body from the substantia nigra (40×). (**T**) p38MAPK deposition in multiple Lewy bodies located in the basal nuclei (80×).

**Figure 5 ijms-23-08376-f005:**
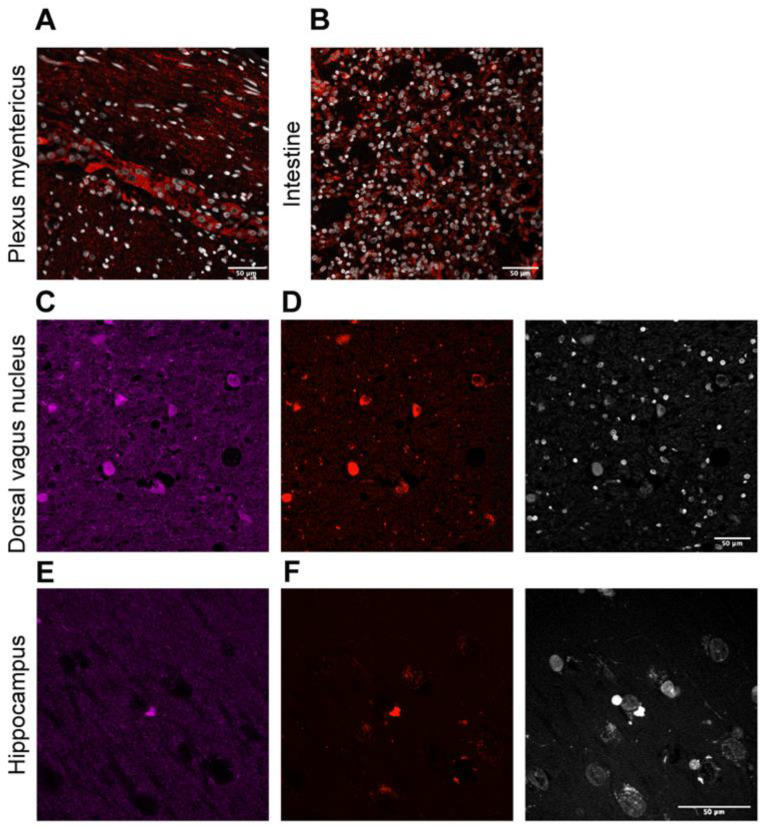
(**A**,**B**) Representative images of norovirus capsid protein VP1 immunofluorescence (red) counterstained with DAPI (blue) revealed abundant norovirus in the myenteric plexus (**A**) and the intestine (**B**). (**C**–**F**) Representative image of double immunofluorescence of VP1 (red) and alpha-synuclein (magenta) in different regions of the brain. There is colocalization in the dorsal vagus nucleus (**C**,**D**) and hippocampus (**E**,**F**). Scale bar: 50 µm.

**Table 1 ijms-23-08376-t001:** Consensus criteria for the diagnosis of Lewy disease according to the LPC system.

Category of Lewy Pathology	Olfactory Bulb	Amygdala	Dorsal Vagal Nucleus and Solitary Nucleus	Medial Temporal Lobe or Cingulate Cortex	Frontal or Parietal Cortex
Olfactory only	+	-	-	-	-
Amygdala predominant	-/+	+	-	-	-
Brainstem predominant	-/+	-/+	+	-	-
Limbic	-/+	-/+	-/+	+	-
Neocortical	-/+	-/+	-/+	-/+	+

## Data Availability

All data generated or analyzed during this study are included in this published article.

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
