# Peer review of "Norovirus Intestinal Infection and Lewy Body Disease in an Older Patient with Acute Cognitive Impairment"

_ijms, 2022, doi:10.3390/ijms23158376_

Round 1

Reviewer 1 Report

In this paper, Moreno-Valladares and colleagues describe a case report of a concomitant norovirus infection along with alpha-synuclein in the gastrointestinal mucosa and Lewy pathology in the CNS of an older woman.

Overall, the manuscript is well written and interesting. In the Introduction, the authors discuss evidence regarding the role of alpha-synuclein as a double-edged sword that not only facilitates cell survival but can also be toxic, leading to its demise. Within this frame, it would be helpful for the reader to add some review papers which discuss the detrimental and beneficial effects of alpha-synuclein and its opposite outcomes in cell homeostasis.

- Tan LY, Tang KH, Lim LYY, Ong JX, Park H, Jung S. α-Synuclein at the Presynaptic Axon Terminal as a Double-Edged Sword. Biomolecules. 2022 Mar 27;12(4):507. doi: 10.3390/biom12040507. PMID: 35454096; PMCID: PMC9029495.

- Sulzer D, Edwards RH. The physiological role of α-synuclein and its relationship to Parkinson's Disease. J Neurochem. 2019 Sep;150(5):475-486. doi: 10.1111/jnc.14810. Epub 2019 Jul 28. PMID: 31269263; PMCID: PMC6707892.

- Ryskalin L, Busceti CL, Limanaqi F, Biagioni F, Gambardella S, Fornai F. A Focus on the Beneficial Effects of Alpha Synuclein and a Re-Appraisal of Synucleinopathies. Curr Protein Pept Sci. 2018;19(6):598-611. doi: 10.2174/1389203718666171117110028. PMID: 29150919; PMCID: PMC5925871.

- Bernal-Conde LD, Ramos-Acevedo R, Reyes-Hernández MA, Balbuena-Olvera AJ, Morales-Moreno ID, Argüero-Sánchez R, Schüle B, Guerra-Crespo M. Alpha-Synuclein Physiology and Pathology: A Perspective on Cellular Structures and Organelles. Front Neurosci. 2020 Jan 23;13:1399. doi: 10.3389/fnins.2019.01399. PMID: 32038126; PMCID: PMC6989544.

Within this frame, it would be interesting to know if the Authors have considered extending the analysis on other alpha-synuclein isoforms, such as proteinase K-resistant protein isoform, or by means of other anti-alpha synuclein primary antibodies beyond BioLegend, Ref.: 807701. This would be interesting since there are data reported in the literature that demonstrate a differential alpha-synuclein immunostaining due to different specificity of currently available primary antibodies.

Minor point:

The Authors should improve the quality of Figure 2 (C-F), Figures 3-5.

Author Response

In this paper, Moreno-Valladares and colleagues describe a case report of a concomitant norovirus infection along with alpha-synuclein in the gastrointestinal mucosa and Lewy pathology in the CNS of an older woman.

Overall, the manuscript is well written and interesting.

AUTHORS: We thank the reviewer for the positive evaluation of our manuscript and for considering it well written and interesting.

In the Introduction, the authors discuss evidence regarding the role of alpha-synuclein as a double-edged sword that not only facilitates cell survival but can also be toxic, leading to its demise. Within this frame, it would be helpful for the reader to add some review papers which discuss the detrimental and beneficial effects of alpha-synuclein and its opposite outcomes in cell homeostasis.

- Tan LY, Tang KH, Lim LYY, Ong JX, Park H, Jung S. α-Synuclein at the Presynaptic Axon Terminal as a Double-Edged Sword. Biomolecules. 2022 Mar 27;12(4):507. doi: 10.3390/biom12040507. PMID: 35454096; PMCID: PMC9029495.

- Sulzer D, Edwards RH. The physiological role of α-synuclein and its relationship to Parkinson's Disease. J Neurochem. 2019 Sep;150(5):475-486. doi: 10.1111/jnc.14810. Epub 2019 Jul 28. PMID: 31269263; PMCID: PMC6707892.

- Ryskalin L, Busceti CL, Limanaqi F, Biagioni F, Gambardella S, Fornai F. A Focus on the Beneficial Effects of Alpha Synuclein and a Re-Appraisal of Synucleinopathies. Curr Protein Pept Sci. 2018;19(6):598-611. doi: 10.2174/1389203718666171117110028. PMID: 29150919; PMCID: PMC5925871.

- Bernal-Conde LD, Ramos-Acevedo R, Reyes-Hernández MA, Balbuena-Olvera AJ, Morales-Moreno ID, Argüero-Sánchez R, Schüle B, Guerra-Crespo M. Alpha-Synuclein Physiology and Pathology: A Perspective on Cellular Structures and Organelles. Front Neurosci. 2020 Jan 23;13:1399. doi: 10.3389/fnins.2019.01399. PMID: 32038126; PMCID: PMC6989544.

AUTHORS: We thank the reviewer for this suggestion. We agree with the double-edge sword role of α-Synuclein and we have added additional references in the manuscript to explain this issue.

Within this frame, it would be interesting to know if the Authors have considered extending the analysis on other alpha-synuclein isoforms, such as proteinase K-resistant protein isoform, or by means of other anti-alpha synuclein primary antibodies beyond BioLegend, Ref.: 807701. This would be interesting since there are data reported in the literature that demonstrate a differential alpha-synuclein immunostaining due to different specificity of currently available primary antibodies.

AUTHORS: We have only used the antibody referred to in the methodology. Although it is true that the sensitivity and specificity could change between antibodies, the used antibody has been extensively used in the bibliography and the microscopic examination carried out have shown LBs in the hematoxylin-eosin, as can be seen in the figures, which excludes the possibility of false positives. On the other hand, the possibility of cross-positive staining has also been commented on, as occurs in the case of smooth muscle, and for which we also offer a bibliographical reference. Regarding additional isoforms, although interesting, we did not tested them since we respectfully consider that this does not fit with the main objective of the case report.

Minor point:

The Authors should improve the quality of Figure 2 (C-F), Figures 3-5.

AUTHORS: We improved the quality of figures.

Reviewer 2 Report

The manuscript described a case report study. here below I reported my major doubts, comments and suggestions:

-Minor comments:

1)substitute "The guardians of the patient" with "the patient's caregivers"

2) Figure 5 needs to be modified as figures can be better distributed and enlarged. 

-Major concern.

I am wondering about the relevance of this work to the international journal of molecular sciences.  I suggest author to submit this manuscript to journals specialising in clinical neurological studies.

Author Response

The manuscript described a case report study. here below I reported my major doubts, comments and suggestions:

-Minor comments:

1)substitute "The guardians of the patient" with "the patient's caregivers"

AUTHORS: The text has been changed according to reviewer´s suggestion.

2) Figure 5 needs to be modified as figures can be better distributed and enlarged. 

AUTHORS: We have changed and tried to distribute better the main figures of the manuscript.

-Major concern.

I am wondering about the relevance of this work to the international journal of molecular sciences.  I suggest author to submit this manuscript to journals specialising in clinical neurological studies.

AUTHORS: We thank reviewer suggestion, however, we respectfully consider that it fits well in the scope of the IJMS journal. We think that rather than clinical aspect (which it does not have any specific syndrome, since the patient was not diagnosed clinically with PD), it is more relevant the molecular aspect of showing the co-existence of pathology associated to norovirus and alpha-synuclein, which allow us to support Braak´s hypothesis of intestinal origin of pathology with LBs.

Reviewer 3 Report

After reading the manuscript entitled "Norovirus intestinal infection and Lewy body disease in an older patient with acute cognitive impairment", I consider it appropriate to be published in IJMS.

The described clinical case is very important because the presented results support Braak's hypothesis about the pathogenic mechanisms underlying synucleinopathies. The abstract of the manuscript clearly explains why this case report is important. The introduction section includes background information connected with the described clinical case in this study. The methods presented in the study have been well described. The case presentation - this part of the manuscript includes all needed information about the patient. The photographic documentation with descriptions is a very valuable part of the reviewed article. In the discussion section, the authors confront their observations with the findings of other researchers.

Minor corrections in the manuscript text must be performed to increase its quality:
line 15 is:... heath.., should be:.. health..
line 36 is:.. eosynophilic.., should be:.. eosinophilic..
line 39 is:.. chromosome.., should be:.. chromosomes..
line 48 is:.. Initial.., should be:.. The initial..
line 57 is:.. formation.., should be:..the formation..
line 69 is:.. immunodepressed.., should be:.. immunosuppressed..
line 78 is:.. or.., should be:..and..
line 78 is:.. based in.., should be:.. based on..
line 106 is:.. amoxycillin.., should be:.. amoxicillin..
line 122 is:.. at.., should be:..in..
line 129 is:.. in.., should be:..on..

Author Response

After reading the manuscript entitled "Norovirus intestinal infection and Lewy body disease in an older patient with acute cognitive impairment", I consider it appropriate to be published in IJMS.

The described clinical case is very important because the presented results support Braak's hypothesis about the pathogenic mechanisms underlying synucleinopathies. The abstract of the manuscript clearly explains why this case report is important. The introduction section includes background information connected with the described clinical case in this study. The methods presented in the study have been well described. The case presentation - this part of the manuscript includes all needed information about the patient. The photographic documentation with descriptions is a very valuable part of the reviewed article. In the discussion section, the authors confront their observations with the findings of other researchers.

AUTHORS : We thank the reviewer for the positive evaluation of our manuscript and for considering appropriate for publication in IJMS.

Minor corrections in the manuscript text must be performed to increase its quality:

line 15 is:... heath.., should be:.. health..

line 36 is:.. eosynophilic.., should be:.. eosinophilic..

line 39 is:.. chromosome.., should be:.. chromosomes..

line 48 is:.. Initial.., should be:.. The initial..

line 57 is:.. formation.., should be:..the formation..

line 69 is:.. immunodepressed.., should be:.. immunosuppressed..

line 78 is:.. or.., should be:..and..

line 78 is:.. based in.., should be:.. based on..

line 106 is:.. amoxycillin.., should be:.. amoxicillin..

line 122 is:.. at.., should be:..in..

line 129 is:.. in.., should be:..on..

AUTHORS : These modifications have been included in the revised version of the manuscript.

Reviewer 4 Report

This is an interesting case study, building on previous observations that gastrointestinal infections with norovirus might be associated with synuclein pathology. Here, the authors present the case of a 76-year-old female, with cognitive symptoms and postmortem detection of synuclein deposits in the gastrointestinal tract, as well as widespread detection of synuclein deposits in the peripheral and central nervous system. Striking is the colocalization of synuclein with norovirus capsid protein in several brain areas, including the dorsal vagal nerve and the hippocampus, which – to the best of my knowledge - has not been reported before.

I only have minor questions/suggestions:

A hypothesis is that infections in the gut (or other peripheral sites) could act as triggers of synucleinopathy – which is discussed in the introduction and discussion. I believe it would be prudent to mention that other types of PD can originate in the brain (‘brain-to-body’), as opposed to cases where the disease might start in the periphery (‘body-to-brain’) – or, alternatively, that PD can be precipitated because of infection (the authors note sudden cognitive deterioration after infection). The study now focuses quite strongly on the Braak hypothesis, but I am not convinced that this case provide strong support of this hypothesis; there might be alternative ways via which the disease might develop or via which infection could impact the CNS. The patient reported intestinal motility disorder between 2016 and 2018, which is only 4-6 years ago, and therefore probably not long enough for synuclein assemblies to transmit via the vagal nerve to hippocampal or cortical areas and cause symptoms. From this case it almost seems that cognitive symptoms were triggered acutely after or even during GI disturbances.

Could the authors discuss the mechanisms related to norovirus infection and how it might potentially act as a neurotropic agent? Some case reports have also suggested norovirus-related encephalitis; the detection of norovirus protein in the hippocampus of the presented case might suggest a mechanism related to systemic infection or infection across vascular epithelial cells? Can the virus also spread retrogradely via the vagal nerve?

In the discussion is often mentioned that synuclein is secreted after norovirus infection? I have looked for the original reference but can’t seem to find what is meant by this. Do the authors mean that synuclein is actually secreted? Or maybe – that it is ‘detected’ outside neurons in peripheral tissue after norovirus infection?

Is there detection of synuclein-positive cells in the absence of norovirus protein? From the images it seems that in affected neurons both proteins are always present together.

Is any of the synuclein in the gut also aggregated, aside from phosphorylated, e.g. are there ThioT/S- or LCO (pFTAA/hFTAA)-positive synuclein inclusions?

There is strong staining with LB509 in the heart muscle. Is this specific staining?

Author Response

This is an interesting case study, building on previous observations that gastrointestinal infections with norovirus might be associated with synuclein pathology. Here, the authors present the case of a 76-year-old female, with cognitive symptoms and postmortem detection of synuclein deposits in the gastrointestinal tract, as well as widespread detection of synuclein deposits in the peripheral and central nervous system. Striking is the colocalization of synuclein with norovirus capsid protein in several brain areas, including the dorsal vagal nerve and the hippocampus, which – to the best of my knowledge - has not been reported before.

AUTHORS: We thank the reviewer for the detailed summary and the positive evaluation of our work 

I only have minor questions/suggestions:

A hypothesis is that infections in the gut (or other peripheral sites) could act as triggers of synucleinopathy – which is discussed in the introduction and discussion. I believe it would be prudent to mention that other types of PD can originate in the brain (‘brain-to-body’), as opposed to cases where the disease might start in the periphery (‘body-to-brain’) – or, alternatively, that PD can be precipitated because of infection (the authors note sudden cognitive deterioration after infection). The study now focuses quite strongly on the Braak hypothesis, but I am not convinced that this case provide strong support of this hypothesis; there might be alternative ways via which the disease might develop or via which infection could impact the CNS. The patient reported intestinal motility disorder between 2016 and 2018, which is only 4-6 years ago, and therefore probably not long enough for synuclein assemblies to transmit via the vagal nerve to hippocampal or cortical areas and cause symptoms. From this case it almost seems that cognitive symptoms were triggered acutely after or even during GI disturbances.

Could the authors discuss the mechanisms related to norovirus infection and how it might potentially act as a neurotropic agent? Some case reports have also suggested norovirus-related encephalitis; the detection of norovirus protein in the hippocampus of the presented case might suggest a mechanism related to systemic infection or infection across vascular epithelial cells? Can the virus also spread retrogradely via the vagal nerve?

AUTHORS: Our case report does not pretend to establish causal relationship and it is intended to illustrate a specific case in which a norovirus infection and pathology by LBs coexist. It could be plausible that the LBs pathology is primary of the CNS, and has been accelerated by the intercurrence of viral infection by norovirus, which is known to be associated with the secretion of Alpha-synuclein at the intestinal level. It would also be plausible, therefore, that the process began in the intestine and accelerated in this last phase of the process. The observation of viral particles forming part of the LBs in the CNS suggests that they have reached the neuronal soma traveling through the nerve fibers, since if they had done so by hematogenous spread, crossing the BBB, they should also be observed outside the cell soma, with inflammatory lesions of the encephalitic type, perhaps necrosis and also foci of accumulation of macrophages and lymphocytes, which are not seen in our case. We have added information in the discussion following reviewer suggestion.

In the discussion is often mentioned that synuclein is secreted after norovirus infection? I have looked for the original reference but can’t seem to find what is meant by this. Do the authors mean that synuclein is actually secreted? Or maybe – that it is ‘detected’ outside neurons in peripheral tissue after norovirus infection?

AUTHORS: The reviewer is correct. We wanted to express that synuclein is detected. We have changed this in the revised version

Is there detection of synuclein-positive cells in the absence of norovirus protein? From the images it seems that in affected neurons both proteins are always present together.

AUTHORS: No, all the images that we have taken indicate that both proteins are detected at the same time.

Is any of the synuclein in the gut also aggregated, aside from phosphorylated, e.g. are there ThioT/S- or LCO (pFTAA/hFTAA)-positive synuclein inclusions?

AUTHORS: This is an interesting question that we have not tested.

There is strong staining with LB509 in the heart muscle. Is this specific staining?

AUTHORS: In this casethe possibility of cross-positive staining is high and we do not consider that this is specific staining. There is a bibliographical reference for this that is part of the manuscript.  

Round 2

Reviewer 2 Report

Despite authors attempt to reply to reviewer's comments, the quality of the manuscript is not satisfactory. The figures are still of low quality and should be improved in representation and distribution.

With data presented by the authors, the possible contributive link between norovirus infection and DLB can only be assumed but not proven. 

I suggest authors to revise the manuscript carefully and to re-submit to another journal rather than IJMS, more  focused on the clinic